# Guidelines for a Finite Element Based Design of Timber Structures and Their Exemplary Application on Modelling of Beech LVL

Janusch Töpler *, Lea Buchholz, Julian Lukas and Ulrike Kuhlmann

Institute of Structural Design, University of Stuttgart, 70569 Stuttgart, Germany
* Correspondence: janusch.toepler@ke.uni-stuttgart.de

**Abstract:** Design verifications of buildings are usually carried out supported by a finite element analysis (FEA), for which, however, there are only a few and almost exclusively non-binding application rules. Within the Cluster of Excellence *Integrative Computational Design and Construction for Architecture* (IntCDC) at the University of Stuttgart, *Guidelines for a Finite Element-Based Design of Timber Structures* have been developed. The scope of the guidelines is daily engineering practice, expert engineering applications and product development and certification. Essential parts of the guidelines are design procedures, modelling (including geometrical, material and imperfection modelling), analysis, model verification and validation and design. The content and application of the guidelines are described and illustrated in this paper using two benchmarks. These two benchmarks, which are based on experimental investigations, deal with the elastic material modelling of glulam made of beech laminated veneer lumber (beech LVL) and dowel-type connections for beech LVL members. The experimental basis of the benchmarks is described. With the experiments for the benchmarks, all Poisson's ratios and the complete elastic material stiffness matrix of beech LVL are determined by means of an optical measuring system. The experimentally determined stiffnesses of the investigated dowel-type connections in beech LVL are compared with normative values. Based on the experiments, a numerical model is developed in RFEM (Dlubal).

**Keywords:** finite element based design; experiments; beech LVL; material properties; Poisson's ratios; connection stiffness; steel-timber dowel-type connections; beam-on-foundation model; embedment behaviour; optical measurement system

## 1. Introduction

### 1.1. General

Computer-aided computations and design using the finite element method (FEM) are nowadays widespread in building practice and science and will continue to gain importance.

However, in building practice there are no regulations on how these numerical calculation methods are to be applied, and so each user has different requirements for numerical models. Furthermore, the application in building practice with the determination of internal forces, stresses and deformations is usually limited to the action side, while the resistances are determined using "manual" design methods described in the standards.

At the Institute of Structural Design at the University of Stuttgart, therefore, *Guidelines for a Finite Element Based Design of Timber Structures* [1] were developed. The research is part of the Cluster of Excellence *Integrative Computational Design in Architecture and Construction* (IntCDC) at the University of Stuttgart [2]. The scope of the guidelines is daily engineering practice, expert engineering application and product development and certification. Additionally, hints for the application of FEM in research are given. In the guidelines, the criteria for the verification and validation of numerical models are defined. Procedures for FE based design within the safety concept of the Eurocodes are given, including methods for the direct numerical computation of structural resistances. Similar

to EN 1990 Annex D "Design assisted by testing" [3], these procedures allow for a design of timber structures beyond the standard design cases of the Eurocodes and enable structural engineers to design innovative constructions and to verify them in the framework of the Eurocodes using the FEM. The guidelines were developed on the basis of prEN 1993-1-14: Eurocode 3: Design of steel structures—Part 1-14: Design assisted by finite element analysis [4].

This paper presents the main content of the developed *Guidelines for a Finite Element Based Design of Timber Structures* [1] and its application by means of two benchmarks on glulam made of beech laminated veneer lumber (beech LVL) GL75 according to ETA-14/0354 [5]. The benchmarks are validated by experiments, which additionally provide important input values for the modelling of beech LVL.

The first benchmark deals with the elastic material behaviour of beech LVL. The bases of this benchmark are experimental investigations of moduli of elasticity (MOE), shear moduli and Poisson's ratios for beech LVL and numerical comparative calculations. The complete anisotropic stiffness matrix for beech LVL GL75 was experimentally determined, and the influence of several load cycles was investigated. The experimentally determined material parameters are used for numerical modelling and recalculation of the tests.

The second benchmark comprises experiments and numerical calculations concerning the load–deformation behaviour of dowel-type connections in beech LVL. Embedment and component tests to determine the connection stiffnesses of dowel-type connections are presented. The results of the embedment tests are used as input values for Beam-on-Foundation (BoF) models, which are utilised to recalculate the component tests.

### 1.2. State of the Art

#### 1.2.1. Finite Element Modelling

The origins of the finite element method (FEM) date back to the 19th century. The first practical application took place in the aerospace industry and was described by Turner et al. [6]. Since then, the calculation method, with which a wide variety of physical problems, e.g., from structural and fluid mechanics, can be described approximately, has developed into the predominant numerical calculation method in the construction sector for stress, strain and deformation analyses.

There is a large amount of literature on the background and application of the FEM (e.g., [7,8]). Since calculation results are strongly dependent on the basic theory, the input values, the modelling, the selected method of analyses, the solver and the results' interpretation, there are various guidelines (e.g., [9–12]) that are intended to ensure the correct application of the FEM. However, experience has shown that these guidelines are rarely applied in building practice in Germany, as they are not binding and the Eurocodes have so far only mentioned numerical methods, but have not described their application. Only EN 1993-1-5 Annex C [13] and EN 1993-1-6 [14] provide application rules for the use of the FEM, but only for the special case of designing steel-plated structures.

Although numerical calculation methods are widely used in practice, there are almost no binding application rules for users in building practice. This leads to a very heterogeneous quality of numerical calculation results in structural design and science, and to uncertainty and unnecessary errors when applying the FEM and to ambiguities in disputes.

A standardisation of the application of the FEM in the design practice is therefore urgently required. In steel construction, such regulations are formulated for the first time in the new prEN 1993-1-14: Eurocode 3: Design of steel structures—Part 1-14: Design assisted by finite element analysis [4].

The definition and implementation of three design methods, which differentiate according to their different design purposes, are fundamental in prEN 1993-1-14 [4]. These are:

- Numerical design calculation requiring a subsequent design check;
- Numerical design calculation with direct resistance check;
- Numerical simulation.

These are described in Section 2.2. These different design purposes do also influence the safety concept. In addition, the terms verification and validation are defined including clear criteria against which numerical models and computational results should be checked in order for the models to be accepted for design verification [4].

There is nothing comparable for timber structures in Europe up to now. The guidelines presented here for the FE based design of timber structures were developed to provide an impulse and a proposal for standardisation of the FEM for timber.

### 1.2.2. Benchmark—Elastic Material Behaviour of Beech LVL

Research activities on glulam made of beech LVL started more than 10 years ago and have focused on the material properties relevant to practice. Dill-Langer and Aicher [15,16] investigated the bending strength and the influence of the size effect for beams up to 2.5 m height. The compressive strength and stiffness were assessed by Blaß and Windeck [17], Dill-Langer and Aicher [16,18], Ehrhart and Steiger and Frangi [19,20] and Kuck [21], but the results were only partly published. Stiffness and plasticising in the grain direction were discussed by Töpler and Kuhlmann [22]. Rolling shear strength and stiffness were studied by Hütter [23], and shear strength and stiffness by Dill-Langer and Aicher [16]. Frese [24] observed the influence of the temperature and compressive stresses in the manufacturing process on the density distribution over the cross-sectional height of the LVL.

In December 2013, the first national technical approval for glulam made of beech LVL, also known as BauBuche GL70, later GL75, was granted [25]. A revised version of this approval was published in 2015 as ETA-14/0354 [5].

The characteristic and average values of the MOE parallel and perpendicular to the grain and the shear modulus in grain direction can be taken from ETA-14/0354 [5]. The Poisson's ratios of beech LVL have not been determined yet.

For the determination of the elastic material stiffness matrix for numerical modelling of beech LVL, there is therefore in particular a lack of test results on Poisson's ratios. No results on MOE perpendicular to the grain and shear moduli have been published yet. This knowledge gap, which limits the numerical modelling of beech LVL, is closed with the first benchmark case.

### 1.2.3. Benchmark—Dowel-Type Connections for Beech LVL

The stiffness calculation of dowel-type connections according to EN 1995-1-1 [26] currently only considers the influence of the timber density and the fastener diameter. However, several investigations (e.g., [27–30]) have shown that there are further important influencing parameters such as the load-to-grain angle, the slenderness of the fastener or the group effect.

In order to analyse the influences on the connections stiffness more precisely and to predict the load–deformation behaviour, Beam-on-Foundation (BoF) models can be used. This modelling approach comprises a beam, representing the fastener, embedded on non-linear springs, which describe the embedment behaviour of the timber and the steel plate. The BoF model for dowel-type connections was first presented by Hochreiner et al. [31], expanded by Schweigler [32] and now implemented in RFEM (Dlubal) by Kuhlmann and Gauß [33,34].

According to Schweigler [32], the embedment stress can be described as a function of the dowel displacement. This analytical approach is shown in Equation (1) and is used in the following to approximate the experimentally determined embedment curves (see also Figure 12a).

$$f_\mathrm{h}(u) = \frac{(k_\mathrm{ser} - k_\mathrm{f}) \cdot u}{\left[1 + \left[\frac{(k_\mathrm{ser} - k_\mathrm{f}) \cdot u}{f_\mathrm{h,int}}\right]^\alpha\right]^{\frac{1}{\alpha}}} + k_\mathrm{f} \cdot u \tag{1}$$

where $f_\mathrm{h}$ is the embedment stress, $u$ the dowel displacement, $k_\mathrm{ser}$ the initial stiffness, $k_\mathrm{f}$ the end gradient of the slip curve (plastic stiffness), $f_\mathrm{h,int}$ the embedment stress at intersection of the tangent from $k_\mathrm{f}$ with the vertical axis and $\alpha$ the transition parameter.

Gradually, the BoF model has become not only of interest in research but also in building practice. By implementing the model in common software, dowel-type connections, even as groups with several fasteners, can be modelled and designed in a simple way considering realistic input parameters, and thus taking into account the actual connection stiffness. Kuhlmann and Gauß pursued this modelling technique, which allows for a more economical design of whole timber structures, within their experimental and numerical investigations on component and embedment tests in the IGF research project No. 20625 N [34] and the Cluster of Excellence IntCDC, RP 7-1 [2]. The mentioned investigations provide the input parameters for the modelling of steel-timber dowel-type connections in beech LVL using a BoF model, which is presented in the second benchmark case.

## 2. Guidelines for a Finite Element Based Design of Timber Structures

### 2.1. General

The *Guidelines for a Finite Element Based Design of Timber Structures* [1] provide regulations for the application of numerical methods for the design of timber structures in ULS and SLS in daily engineering practice and expert engineering applications, for product development and certification and give hints for research. Excerpts from the developed guidelines are presented in the following, including the design procedure and the design methods. For illustrating the application of the guidelines, two benchmarks are discussed in Sections 3 and 4. Further information on modelling, types of analyses, verification and validation, design methodology, documentation, benchmarks, etc., can be found in the guidelines [1]. The development of the guidelines was based on prEN 1993-1-14 [4].

### 2.2. Procedure, Design Methods and Types of Analysis

Figure 1 gives an overview of the procedure for the FE based design. In the first step (1), the **problem** to be investigated is clarified.

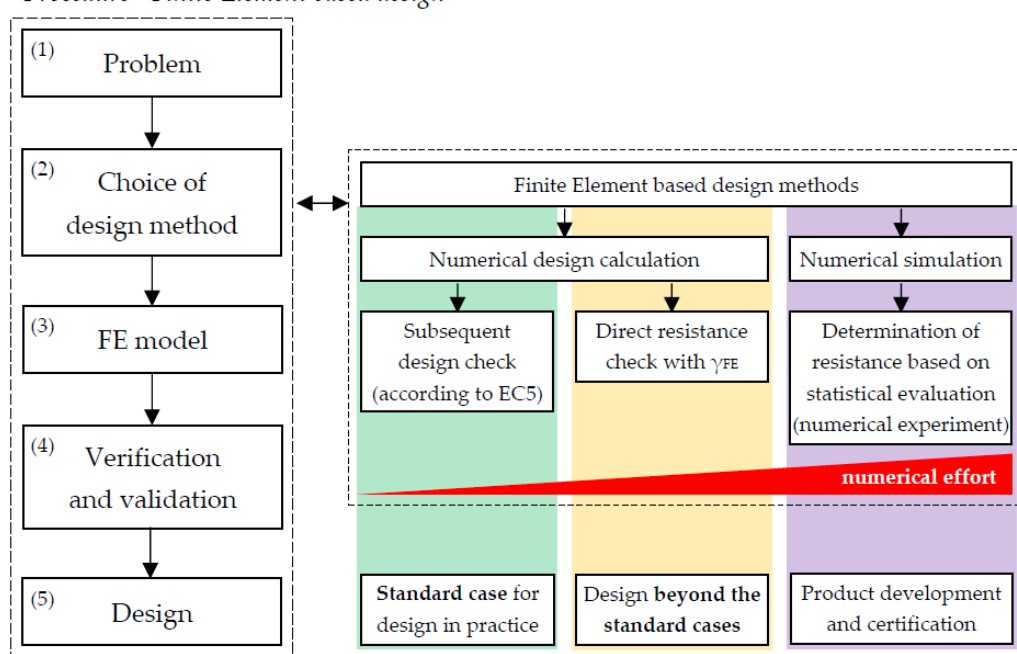

**Figure 1.** Procedure for the FE based design [1].

In the next step (2), the appropriate **design method** is chosen. Three different design methods are defined according to the different design purposes, which are presented in the following and are illustrated schematically in Figure 1.

The **numerical design calculation requiring a subsequent design check** describes the FE based design in structural engineering that is common practice today (daily engineering practice and standard design cases covered by the Eurocodes). Input values of the numerical models are nominal values of the geometry and material parameters according to EN 1995-1-1 [26], relevant product standards or technical approvals. The results are internal forces, stresses, strains and deformations (effects of actions, also referred to as system response quantities (SQRs)), with which the design verifications are carried out according to the design formulas in the current standards (e.g., EN 1995-1-1 [26]). The model uncertainty is generally covered by the partial factor $\gamma_M$ according to EN 1995-1-1 [26], which was calibrated on reference cases.

In the **numerical design calculation with direct resistance check**, the design is carried out on the basis of numerically determined effects of actions <u>and resistances</u>. This expert engineering application may be used, e.g., for design cases which go beyond the standard design cases defined in the Eurocodes. Limiting values may be, besides stress limits, deformation or strain limits, e.g., for bending/compression failure. Input values of the numerical models are nominal values of the geometry and material parameters according to EN 1995-1-1 [26], relevant product standards or technical approvals. The results are internal forces, stresses, strains and deformations (effects of actions or SQRs) and resistances, which can be used for a direct resistance check taking into account the partial factors according to EN 1990 [3], EN 1995-1-1 [26] and the partial factor for modelling $\gamma_{FE}$. The partial factor for modelling $\gamma_{FE}$ covers the uncertainties of the numerical model and the type of analysis considering the difference between the numerical model and physical reality. $\gamma_{FE}$ is independent of all other partial factors given in EN 1990 [3] and all parts of EN 1995. $\gamma_{FE}$ is determined in the validation (see Section 2.3) or assumed as a reasonable value according to the National Annex if no tests or benchmarks are available.

**Numerical simulations** are numerical computations that complement, extend or replace physical experiments, e.g., for product development and certification. They are used to determine the direct resistance of a structure. Input values for geometry and material properties are measured values, mean values or scattering values. Scattering values (their distribution functions, mean values, standard deviations, etc.) should be chosen according to experiments, the literature or experience. The design resistances $f_d$ may be determined by a statistical evaluation of the numerical computation results according to EN 1990 [3] under consideration of the modelling uncertainty.

The three design methods differ in terms of modelling, type of analysis and validation and verification process.

**Types of analysis** are the linear elastic analysis (LA), linear bifurcation (eigenvalue) analysis (LBA), materially non-linear analysis (MNA), geometrically non-linear analysis (GNA), geometrically and materially non-linear analysis (GMNA), geometrically non-linear elastic analysis with imperfections (GNIA) and geometrically and materially non-linear analysis with imperfections (GMNIA).

### 2.3. Verification and Validation

After choosing the design method and the type of analysis and the creation of the numerical model (Figure 1 (3)), the **verification** and the **validation** of the model (Figure 1 (4)) are carried out to prove that the model is appropriate. Figure 2 illustrates a graphical interpretation of the verification and validation process.

The **verification** demonstrates that the numerical model and analysis are properly implemented, understood and applied. Additionally, it demonstrates that the used numerical solution is a good approximation of exact mathematical solutions/mechanical models or benchmarks. The verification includes the following steps:

1.  In the **engineering judgement**, the main calculation results of the FE model (SQRs, which may be internal forces, load–deformation behaviour, . . . ) are checked using simple mechanical models, benchmarks or experience.

2. Within the **discretisation check**, it is shown that the chosen element type and size are accurate for the analysed problem and that the calculation results are not significantly influenced by the discretisation. A mesh convergence study is executed to check if the relevant SRQs converge when the mesh is refined. The chosen mesh size should satisfy the 5% test. If computation times are low, a 1% test may be applied. The application of these tests is described in Section 3.3.2.

3. In a **solver convergence study**, it is ensured that the numerical results have converged when the computation of a load or displacement step is finished.

4. The **sensitivity check** is a variation of the relevant input parameters and determines which parameters are crucial to the relevant SRQ and whether these parameters should be defined with higher precision or not. Geometrical and material properties and the size of time/load steps should be checked.

5. In the examination of **imperfection sensitivity,** whether and which imperfections influence the calculation results are checked (SQRs).

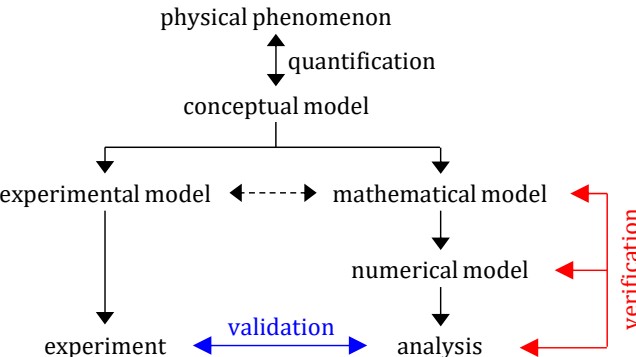

**Figure 2.** Interpretation of a validation and verification process [4].

For standard design cases with existing resistance models in the Eurocodes, the verification steps 2. to 5. may be omitted on the basis of experience.

**Validation** is the comparison of numerical results to known accurate solutions (benchmarks) to demonstrate that the model correctly or conservatively captures the physical phenomena to be modelled. It may result in a quantification of the model uncertainty. Benchmarks can consist of accurate analytical, numerical or experimental results. The differences between the numerical calculation results and the benchmark are evaluated in the validation. Geometry and material parameters are chosen along the benchmark and might therefore differ from the values later used in the design model. The validation procedure depends on the chosen design method.

In the case of numerical design calculations requiring a subsequent design check, the validation may be eliminated, if standard models, for which experience exists, are used for the design and it can be assumed that the model uncertainty is covered by the partial factor $\gamma_M$.

For numerical design calculations with a direct resistance check, the model should be validated, if possible, with the determination of the partial factor for modelling $\gamma_{FE}$. $\gamma_{FE}$ covers the uncertainties of the numerical model and the executed type of analysis and does not override the application of any other partial factors given in the Eurocodes. The partial factor for modelling $\gamma_{FE}$ may be taken as a global factor according to national regulations or according to Equation (2).

$$\gamma_{FE} = \frac{1}{m_x \cdot (1 - k_n \cdot V_x)} \geq 1.0 \tag{2}$$

$$m_x = \text{mean}\left(\frac{f_{test,known} \text{ or } f_{k,known}}{f_{check}}\right) \tag{3}$$

$$V_{\text{x}} = \text{COV}\left( \frac{f_{\text{test,known}} \text{ or } f_{\text{k,known}}}{f_{\text{check}}} \right) \tag{4}$$

where $k_{\text{n}}$ is the characteristic fractile factor according to EN 1990 [3], Annex D, Table D.1 for $V_{\text{x}}$ unknown; $f_{\text{test,known}}$ is the known strength from the test results; $f_{\text{k,known}}$ is the calculated or known characteristic strength; and $f_{\text{check}}$ is the numerically computed strength.

In case of numerical simulations, the validation may be conducted by determining $\gamma_{\text{FE}}$, similar to numerical design calculations with a direct resistance check. Alternatively, if scattering input values are used for geometrical and material properties of the numerical model and a Monte Carlo simulation is conducted, the numerical results can be compared with experimentally determined mean values and scatter, which allows one to determine the model uncertainty. Alternatively, a Monte Carlo simulation with known scatter may lead to a direct partial factor including the model uncertainty following the rules of EN 1990 Annex D "Design assisted by testing" [3].

*2.4. Design*

Based on a verified and validated numerical model, the **design** can be carried out in the last step (Figure 1 (5)). Depending on the selected design method, different procedures can be distinguished.

In the case of a **numerical design calculation requiring a subsequent design check**, the numerically determined SQRs (internal forces, stresses, deformations, strains) are used for design verifications according to the relevant parts of EN 1995. Nominal values are chosen as input values for geometry and material parameters. Partial factors are considered according to EN 1990 [3] and EN 1995-1-1 [26], which also include the influence of the model uncertainty.

In the **numerical design calculations with direct resistance check**, nominal values are chosen as the input for geometry and material parameters. Design loads of the governing loading case combinations are applied in the FE model and multiplied by a load amplification factor. The results are the relevant load–deformation curves and stress–strain curves of the FE model. Figure 3 shows the possible resulting load–deformation/strain curves. The numerically determined structural resistance $f_{\text{FE}}$ results from the minimum of the failure criteria C1, C2 and C3 of all relevant failure mechanisms. C1 is the ultimate stress criterion (e.g., tensile strength $f_{\text{t,0}}$). C2 is the maximum load level of the numerically determined load–deformation curve (e.g., for in-plane buckling of columns). C3 is the load when the largest tolerable strain or deformation is reached. The resistances are determined for each governing failure mechanism (e.g., tension parallel to the grain, rolling shear, deformation criteria, etc.). The characteristic and the design resistances can be calculated according to Equations (5) and (6).

$$f_{\text{k}} = \frac{f_{\text{FE}}}{\gamma_{\text{FE}}} \tag{5}$$

$$f_{\text{d}} = k_{\text{mod}} \frac{f_{\text{k}}}{\gamma_{\text{M}}} \tag{6}$$

where $k_{\text{mod}}$ and $\gamma_{\text{M}}$ are chosen according to EN 1995-1-1 [26]. Further factors such as $k_{\text{h}}$ for the size effect may be considered according to EN 1995-1-1 [26].

In the case of **numerical simulations**, the procedure for determining the structural resistance $f_{\text{FE}}$ for one set of material and geometry parameters is identical to the numerical design calculations with a direct resistance check. In contrast, as the scattering of material and geometry parameters is included in the numerical investigations, the characteristic strength $f_{\text{k}}$ or the design strength $f_{\text{d}}$ are determined based on a statistical evaluation according to EN 1990 under consideration of the modelling uncertainty.

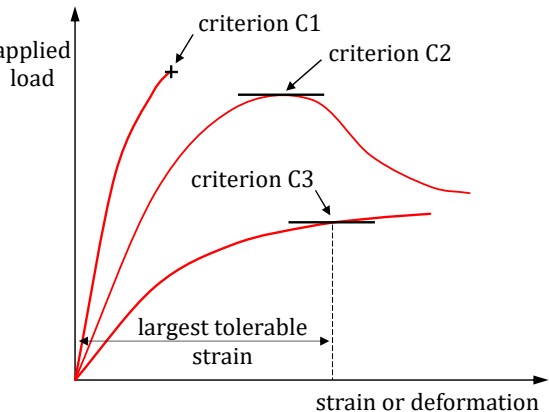

**Figure 3.** Determination of structural resistance by materially non-linear analysis [1].

### 3. Benchmark—Elastic Material Behaviour of Beech LVL

*3.1. General*

Within the research project RP 7-1 of the Cluster of Excellence IntCDC [2] at the University of Stuttgart, experimental investigations were carried out to determine the elastic material properties of beech LVL GL75 [5]. The results allow for more sophisticated numerical modelling of beech LVL and provide important findings for the guidelines.

All entries of the elastic stiffness matrix including Poisson's ratios were determined from bending and compression tests using the optical measuring system ARAMIS Adjustable. In addition, FE modelling for the recalculation of the tests applying the guidelines was conducted. A detailed description of the investigations is given in [35,36].

*3.2. Experimental Investigations*

3.2.1. Test Specimens, Configuration and Execution

The **test programme** comprised 6 test specimens with dimensions height × width × length = 100 mm × 100 mm × 1900 mm made of beech LVL GL75 according to ETA-14/0354 [5].

From the elastic 3-point bending tests, the MOE $E_{\text{L,m,R/T}}$ (indices: L = longitudinal direction, m = bending, R/T = loading in radial/flatwise or tangential/edgewise direction) and the shear moduli $G_{\text{LR/LT}}$ were determined (Figure 4a). Each beam was tested at three different spans $l_s$ = 1800/1200/600 mm and on edgewise/tangential and flatwise/radial loading. Subsequently, 12 test specimens for compression tests parallel to the grain with dimensions height × width × length = 100 mm × 100 mm × 500 mm were cut from the 6 bending test specimens and tested (Figure 4b). Afterwards, the 12 compression test specimens were separated into 100 mm long cubes, and the three middle cubes were rotated, glued together and tested in flatwise/radial direction (Figure 4c). This was repeated to produce the edgewise/tangential compression test specimens from the flatwise/radial compression test specimens. Based on the compression tests, all MOE and Poisson's ratios were determined on the same test specimen.

Following the compression tests, the oven-dry density and wood moisture content (MC) were assessed on two 2 cm thick slices taken from each of the 6 bending test specimens in sections A07 and A13 (Figure 4a, green shading) using the dry oven method from EN 13183-1 [37].

The geometry of the test specimens was measured with a digital calliper and tape measure before the tests were conducted.

The tests were carried out and evaluated based on EN 408 [38]. The load was applied in at least two load cycles, based on EN 26891 [39]. The applied maximum load corresponded to 40% of the estimated mean load-bearing capacity.

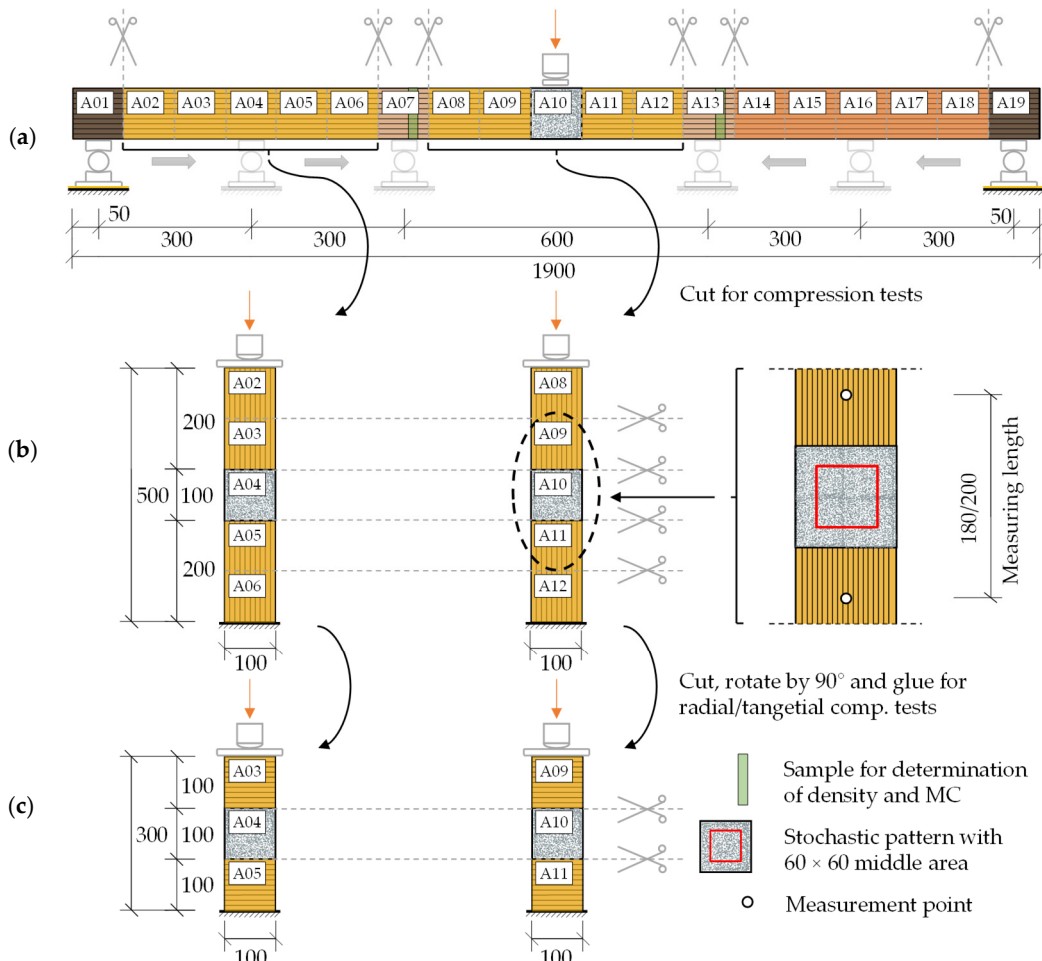

**Figure 4.** Test setup and partitioning of timber test specimens: (**a**) bending tests flatwise/radial and edgewise/tangential; (**b**) compression test longitudinal; (**c**) compression tests flatwise/radial and edgewise/tangential.

The optical measuring systems ARAMIS Adjustable 4M and 12M from GOM GmbH were used. With this, the position of the defined points in space can be measured very precisely in defined time steps by means of digital image correlation. For this purpose, a stochastic pattern was sprayed onto the test specimens (Figure 4). This was supplemented by measurement points attached at specific locations (Figure 4), which were used for determining the MOE under compression according to EN 408 [38]. Deviating from EN 408 [38], a measuring length of 200 mm (instead of 400 mm) was chosen for the determination of the MOE in the grain direction (Figure 4), since a constant stress and strain distribution over the cross-section was found in preliminary numerical investigations for this measuring length. This was confirmed by the test results. The measurements were carried out with two coupled Aramis systems from both sides of the test specimens. Thereby, both sides of the test specimens were measured simultaneously in the bending tests and all four sides in the compression tests.

### 3.2.2. Evaluation and Results

For the **test evaluation**, Figure 5a shows the load–deformation curves of the bending tests with flatwise/radial loading for the three investigated spans $l_s$. Based on these curves, an effective MOE for bending $E_{L,m,R/T,eff}$ was determined with a regression analysis according to EN 408 [38], which included influences from shear and bending (Figure 5b). The MOE for bending $E_{L,m,R/T}$ and shear moduli $G_{LR/LT}$ were determined according to Albers [40] with Equation (7) (determination of the vertical bending and shear deformation

$v$ of rectangular cross-sections at 3-point bending according to Timošenko and Gere [41]) using the experimental results for $l_s$ = 1800 and 600 mm.

$$v = \frac{F \cdot l_s^3}{48 \cdot E_{L,m,R/T,eff} I} = \frac{F \cdot l_s^3}{48 \cdot E_{L,m,R/T} I} + \frac{3 \cdot F \cdot l_s}{10 \cdot G_{LR/LT} A} \tag{7}$$

where $F$ is the total acting point load, $l_s$ is the span, $I$ is the moment of inertia and $A$ is the cross-sectional area.

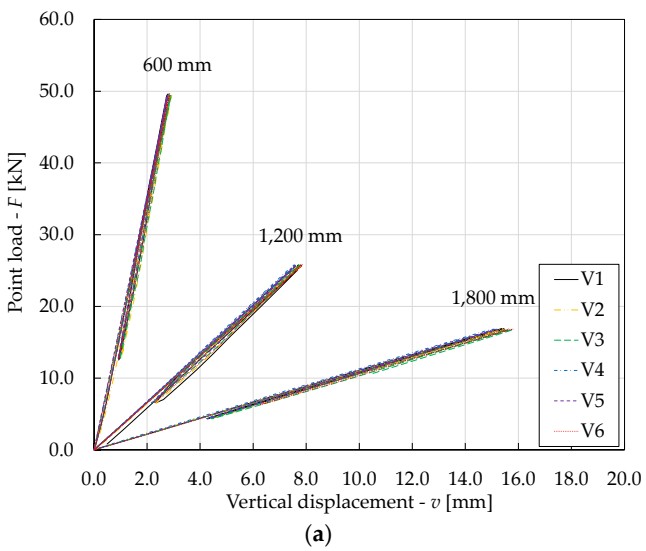

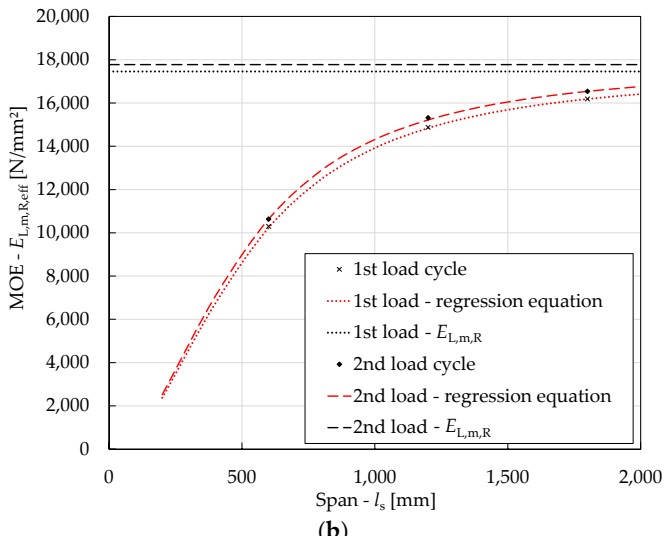

(a)          (b)

**Figure 5.** Results and evaluation of bending tests: (**a**) Load–deformation curves of the bending tests with flatwise/radial orientation of the lamallae for the three investigated spans $l_s$ = 1800/1200/600 mm; (**b**) Evaluation of the bending test BR_V2 with flatwise/radial orientation of the lamallae; effective MOE $E_{L,m,R,eff}$, regression curve and MOE $E_{L,m,R}$ depending on the span $l_s$ for both load cycles.

Figure 5b shows the effective MOE for bending $E_{L,m,R,eff}$ determined from the load–deformation curves (Figure 5a), the regression equations and the determined MOE for bending $E_{L,m,R}$ as a function of the span $l_s$ at the 1st and 2nd load cycle for the test specimen BR_V2.

The MOE from the compression tests $E_{L/R/T}$ was evaluated according to EN 408 [38], whereby the deformation $v$ was taken as the mean value of the change of the measuring length (Figure 4b) at the 4 sides of a test specimen.

The Poisson's ratios $\mu_{LR}$, $\mu_{RL}$, $\mu_{LT}$ and $\mu_{TL}$ were computed by the mean values of the surface strains in the 60 mm × 60 mm middle areas (Figure 4b,c. The Poisson's ratios $\mu_{RT}$ and $\mu_{TR}$ were determined according to Neuhaus [42], on the basis of the distance between the opposite surfaces of the test specimen.

The **results** of all bending and compression tests, the load–deformation curves, were in principle linear (Figure 5a). The slope of the 2nd load cycle always exceeded that of the 1st load cycle by up to 14%. The first bending test specimen was carried out with 6 load cycles. The slopes of the 2nd and all further load cycles differed only slightly from each other (<0.2%). Therefore, all further tests were conducted with two load cycles. For the bending tests, the larger slope of the 2nd loading cycle can partially be attributed to a slip in the test setup during the 1st load cycle. Therefore, the results of the 2nd loading cycle seem to be the ones relevant in practice.

Table 1 lists the mean values and coefficients of variation (COV) of the material properties determined in the experimental investigations of beech LVL GL75 [5] for both load cycles at an average wood moisture content of 5.5%. The MOE along the grain differed depending on the type of loading (bending or compression). The bending MOE $E_{L,m,R/T}$ exceeded the MOE $E_L$ from compression tests by 1.8% to 2.3%. According to Egner [43], the

following usually applies to timber: $E_{\text{tension}} > E_{\text{bending}} > E_{\text{compression}}$, which was confirmed by the experimental results. The scatter of the experimental results of the MOE and shear moduli was quite low with a COV $\leq 7\%$. Due to the manufacturing process, glulam made of beech LVL has only few natural defects such as knots, and these are distributed quite evenly over the cross-section. In addition, all test specimens originated from one production batch.

**Table 1.** Mean values of material properties from bending and compression tests on glulam made of beech laminated veneer lumber GL75 according to ETA-14/0354 [5].

| Material Property | Mean Value (COV) | |
|---|---|---|
| Wood moisture content | 5.54% (1.94%) | |
| Oven-dry density | | |
| $\rho$ | 803 kg/m$^3$ (2.10%) | |
| $\rho_0$ | 780 kg/m$^3$ (2.29%) | |
| | 1st load cycle | 2nd load cycle |
| Moduli of elasticity (MOE) | | |
| $E_L$ | 17,055 N/mm$^2$ (3.76%) | 17,259 N/mm$^2$ (3.55%) |
| $E_{L,m,R}$ | 17,436 N/mm$^2$ (1.68%) | 17,667 N/mm$^2$ (1.87%) |
| $E_{L,m,T}$ | 17,491 N/mm$^2$ (1.75%) | 17,576 N/mm$^2$ (1.12%) |
| $E_R$ | 740 N/mm$^2$ (4.88%) | 840 N/mm$^2$ (4.32%) |
| $E_T$ | 862 N/mm$^2$ (7.09%) | 966 N/mm$^2$ (5.47%) |
| Shear moduli | | |
| $G_{LR}$ | 817 N/mm$^2$ (1.66%) | 909 N/mm$^2$ (4.84%) |
| $G_{LT}$ | 883 N/mm$^2$ (4.59%) | 1006 N/mm$^2$ (6.53%) |
| Poisson's ratios | | |
| $\mu_{LR}$ | 0.3200 (15.7%) | 0.3127 (23.7%) |
| $\mu_{LT}$ | 0.5004 (11.8%) | 0.5167 (13.4%) |
| $\mu_{RT}$ | 0.1992 (13.1%) | 0.1978 (15.1%) |
| $\mu_{RL}$ | 0.0143 (14.3%) | 0.0145 (17.1%) |
| $\mu_{TR}$ | 0.2428 (11.9%) | 0.2506 (13.6%) |
| $\mu_{TL}$ | 0.0246 (14.3%) | 0.0270 (11.2%) |

Based on the material values in Table 1 for the 2nd load cycle, the elastic stiffness matrix **C**, Equation (8), of glulam made of beech LVL GL75 [5] can be calculated for a wood moisture content of 5.5%. The first index indicates the direction of the force, and the second of the deformation.

$$\mathbf{C} = \begin{bmatrix} E_{LL} & E_{LR} & E_{LT} & 0 & 0 & 0 \\ E_{RL} & E_{RR} & E_{RT} & 0 & 0 & 0 \\ E_{TL} & E_{TR} & E_{TT} & 0 & 0 & 0 \\ 0 & 0 & 0 & G_{RT} & 0 & 0 \\ 0 & 0 & 0 & 0 & G_{LT} & 0 \\ 0 & 0 & 0 & 0 & 0 & G_{LR} \end{bmatrix} = \begin{bmatrix} 17,669 & 369 & 570 & 0 & 0 & 0 \\ 410 & 892 & 235 & 0 & 0 & 0 \\ 603 & 214 & 1036 & 0 & 0 & 0 \\ 0 & 0 & 0 & 50 & 0 & 0 \\ 0 & 0 & 0 & 0 & 1006 & 0 \\ 0 & 0 & 0 & 0 & 0 & 909 \end{bmatrix} \tag{8}$$

The maximum difference of $E_{LR}$ to $E_{RL}$, $E_{LT}$ to $E_{TL}$ and $E_{TR}$ to $E_{RT}$ (Equation (8)) was 11%, which could be explained by the material scatter and the measuring accuracy. An asymmetry of the elastic material stiffness matrix of timber found by Neuhaus [42] for spruce cannot fully be ruled out for beech LVL GL75 by the test results.

The shear modulus $G_{RT}$ could not be determined from the tests.

The measured mean width of the bending test specimens (Figure 4a) in the flatwise/radial direction was 98.7 mm and in edgewise/tangential direction was 100.2 mm. The width of the

compression test specimens in the radial and tangential direction (Figure 4c) was reduced to a minimum of 95 mm by the production.

### 3.3. Numerical Investigations

#### 3.3.1. Modelling

FE solid models were developed for the recalculation of each compression and bending test (see Section 3.2.1 and Figure 4a–c) and for demonstrating the application of the *Guidelines for a Finite Element Based Design of Timber Structures* [1]. This is **step (1)** of the design procedure described in Figure 1. Exemplary, the numerical model of a bending test with span $l_s$ = 1800 mm is illustrated in Figure 6.

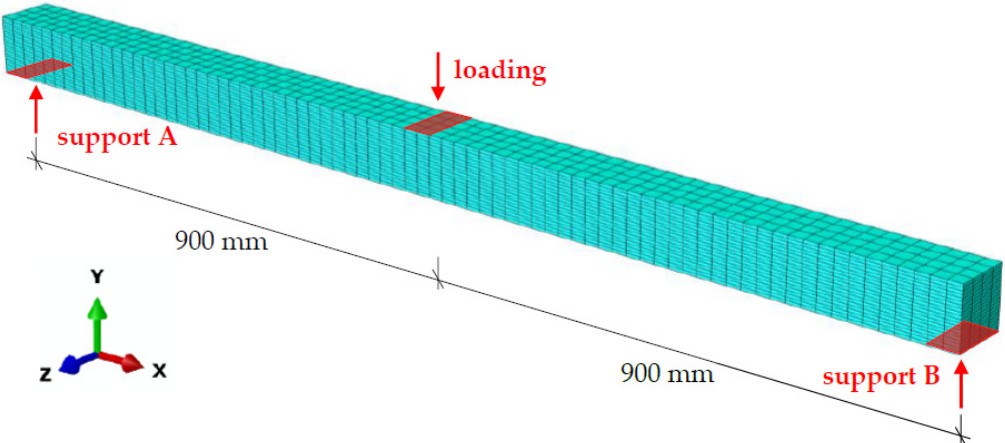

**Figure 6.** FE solid model of a bending test with span 1800 mm including support and loading areas.

The design method **numerical design calculation with direct resistance check** was chosen, **step (2)**. This method was selected to illustrate how the partial factor for modelling $\gamma_{FE}$ may be calculated. The computations were carried out with Abaqus/CAE 2018.

For numerical modelling of the tests, **step (3)**, according to the chosen design method, measured input values were used for the geometry and the material parameters for verification and validation (Section 3.3.2) and nominal values for design (Section 3.3.3).

The boundary conditions and loading were modelled as realistically as possible according to the tests. For example, for the bending tests (see Figure 6), two area supports A and B which additionally restrict rotations around the *x*-axis (fork bearings) and allow for free rotation around the *y*- and *z*-axis were defined. The load was applied as a distributed load at midspan. The dead weight of the members was neglected as it was less than 0.5% of the maximum load.

An ideal elastic material behaviour was assumed, whereby, for verification and validation, the entries of the material stiffness matrix were determined on the basis of the test results for each test specimen (see, e.g., mean values in Equation (8)).

Material non-linearities were neglected, as the load did not exceed 40% of the estimated load-bearing capacity ensuring linear elastic material behaviour [22].

The 20-node reduced integrated volume elements (C3D20R) with the mesh size given in Table 2 were chosen. As the calculation procedure type, *static*, *general* with *direct sparse solver*, *full newton* and one load increment equal to 100% of the maximum load were chosen (fully elastic behaviour).

**Table 2.** Number of elements of the FE mesh of the bending and the compression tests in the respective direction.

|  | No. Over Length | No. Over Width | No. Over Height |
|---|---|---|---|
| Bending tests | 76 | 4 | 20 |
| Compression tests | 50 | 10 | 10 |

3.3.2. Verification and Validation

The verification is illustrated here for the bending test BR_V1_600 mm and the validation for all bending tests, **step (4)** in Figure 1. Material and geometric parameters were taken according to the test results and measurements.

For the **verification,** the following steps were carried out. Results are given for a single load at midspan of 31.24 kN:

1.  **Engineering judgement**: The longitudinal stresses $\sigma_x$ and deformations $v$ displayed in Figure 7a,b is plausible and correspond well to experience. By hand calculation, bending stresses $\sigma_x$ according to Equation (9) and vertical deformations $v$ according to Equation (10) can be determined.

$$\sigma_x = \frac{F \cdot l_s}{4 \cdot W} = 28.32 \text{ N/mm}^2 \tag{9}$$

$$v = \frac{F \cdot l_s{}^3}{48 \cdot EI} + \frac{3 \cdot F \cdot l_s}{10 \cdot GA} = 1.546 \text{ mm} \tag{10}$$

where $F$ = 31,240 N; $l_s$ = 600 mm; $H$ = 100.3 mm; $B$ = 98.7 mm; $E$ = 17,566 N/mm$^2$; $G$ = 976 N/mm$^2$.

2.  **Discretisation check:** Figure 7c,d illustrate the influence of the number of elements over the beam height on the maximum longitudinal stresses $\sigma_x$ and the vertical deformations $v$. To determine the required mesh size, a regression line was used to estimate the correct value of the SQR on the $y$-axis. A mesh that yields results with a maximum deviation of 5% from this target value is sufficient (5% tests). Due to short computation times and for better accuracy, a difference of 1% to the target value was aimed for here (1% tests). The mesh size over the length and width was investigated in the same manner. The selected element type and the element shape were also checked.

3.  **Solver convergence study:** The residual force and moment were studied, which both decreased from iteration to iteration until they were smaller than the default convergence criteria in Abaqus.

4.  **Sensitivity check:** Figure 7e,f show the influence of the variation of the shear moduli $G_{RL}$ and $G_{RT}$ on the maximum longitudinal stress $\sigma_x$ for BR_V1_600 mm. While $G_{RL}$ had a significant influence on the results and should therefore be specified as accurately as possible, the influence of $G_{RT}$ on $\sigma_x$ was negligible, which justifies the assumption $G_{RT}$ = 50 N/mm$^2$. The same studies were carried out for all essential input parameters.

5.  The assessment of the **imperfection sensitivity** could be omitted for the FE models of the compression and bending tests, as there was no influence of imperfections to be expected due to the compact shape of the test specimens.

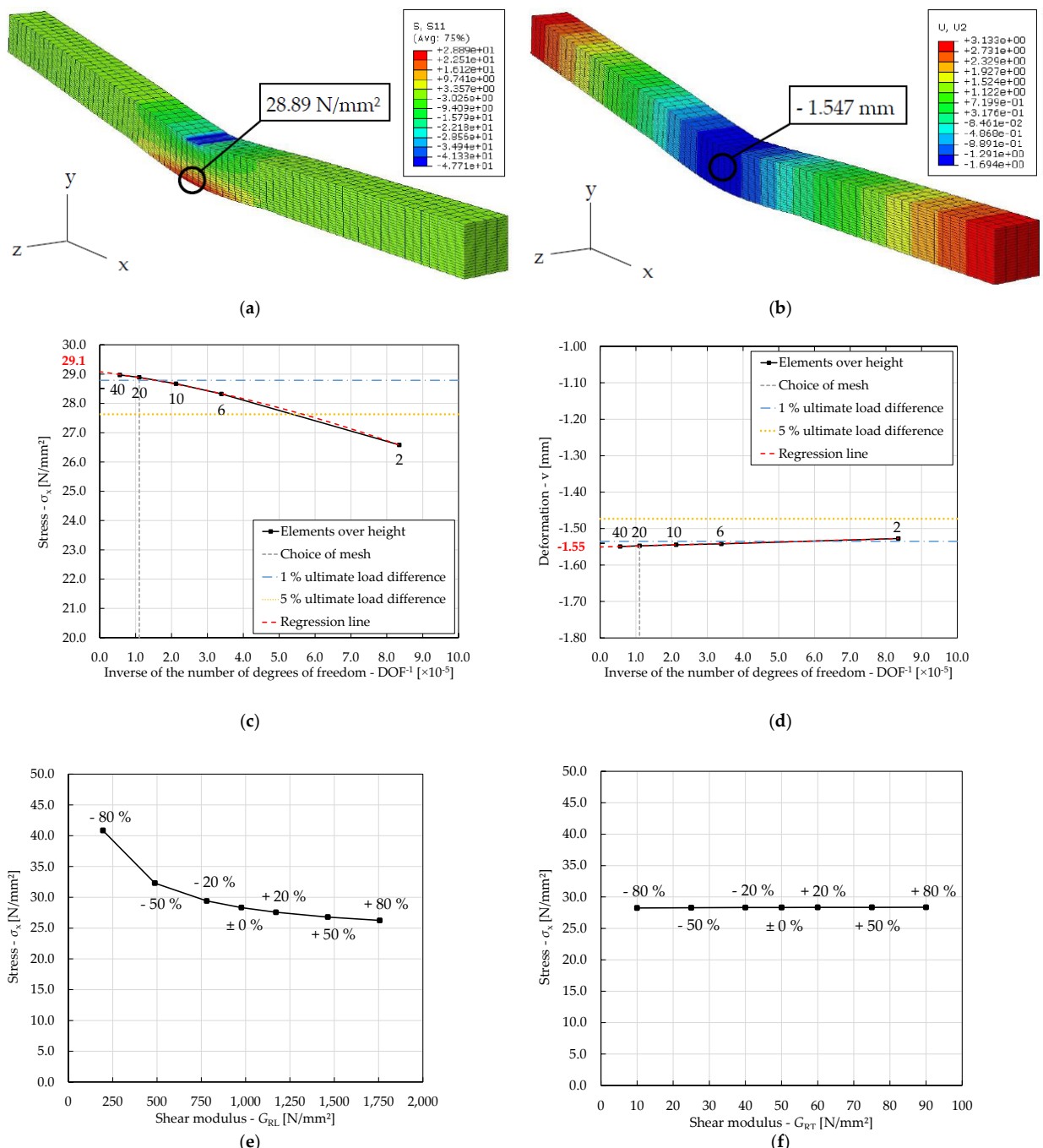

**Figure 7.** Model verification: (**a**) Longitudinal stress $\sigma_x$ (S11), BR_V1_600 mm; (**b**) Deformation $v$ (U2, y-direction), BR_V1_600 mm; (**c**) Numerically determined maximum longitudinal stresses $\sigma_x$ when varying the number of elements over the height (plotted over the inverse of the number of degrees of freedom DOF$^{-1}$) for BR_V1_600 mm; (**d**) Numerically determined maximum displacements $v$ (y-direction) when varying the number of elements over the height (plotted over the inverse of the number of degrees of freedom DOF$^{-1}$) for BR_V1_600 mm; (**e**) Numerically determined maximum longitudinal stresses $\sigma_x$ for variations of the shear modulus $G_{RL}$ for BR_V1_600 mm; (**f**) Numerically determined maximum longitudinal stresses $\sigma_x$ for variations of the shear modulus $G_{RT}$ for BR_V1_600 mm.

For the **validation**, an excerpt of the effective bending MOE $E_{L,m,R/T,eff}$ of 6 of the 36 bending tests is given in Table 3. The experimentally and numerically determined values of the effective bending MOE deviated from each other by a maximum of 2%. Across all $E_{test,known}/E_{check}$, the mean value was $m_x = 1.003$ and the coefficient of variation was

$V_x = 0.006$ (see Equations (2)–(4)). With 36 tests, $k_n$ according to EN 1990 Annex D [3] was $k_n = 1.73$. $\gamma_{FE}$ can thus be calculated from Equation (2) to be $\gamma_{FE} = 1.01$.

**Table 3.** Experimentally and numerically determined values $E_{L,m,R,eff}$ of bending test specimens with $l_s = 600$ mm.

|  | BR_V1 [N/mm²] | BR_V2 [N/mm²] | BR_V3 [N/mm²] | BR_V4 [N/mm²] | BR_V5 [N/mm²] | BR_V6 [N/mm²] |
|---|---|---|---|---|---|---|
| Experimental $E_{test,known}$ | 10,958 | 10,633 | 10,661 | 10,636 | 10,707 | 10,611 |
| Numerical $E_{check}$ | 10,919 | 10,685 | 10,611 | 10,597 | 10,703 | 10,558 |

The FE model was thus verified and validated for non-imperfection-sensitive timber beams under bending stress in the range of elastic deformations.

### 3.3.3. Design

The **design** for the serviceability limit state (SLS) was executed and the corresponding resistance $R_k$ determined, as **step (5)** in Figure 1.

The computation was carried out for an 1800 mm spanning beam with a concentrated load at midspan (Figures 4a and 6) with nominal cross-sectional dimensions 1900 mm × 100 mm × 100 mm. The characteristic (nominal) material values according to ETA-14/0354 [5] were used, supplemented by the experimentally determined values of the Poisson's ratios from Table 1 (Table 4). $E_{R/T}$ was thus underestimated, which, however, had no significant influence on the results according to the sensitivity check.

**Table 4.** Nominal material values for an FE based design calculation of beech LVL [5].

| $E_L$ [N/mm²] | $E_{R/T}$ [N/mm²] | $G_{LR/LT}$ [N/mm²] | $G_{RT}$ [N/mm²] | $\mu_{LR}$ [-] | $\mu_{RL}$ [-] | $\mu_{LT}$ [-] | $\mu_{TL}$ [-] | $\mu_{RT}$ [-] | $\mu_{TR}$ [-] |
|---|---|---|---|---|---|---|---|---|---|
| 15,300 | 400 | 760 | 50 | 0.3127 | 0.0145 | 0.5167 | 0.0270 | 0.1978 | 0.2506 |

EN 1995-1-1 [26] suggests a deformation limit of $L/300 = 1800/300 = 6$ mm. The loading in the FE model was therefore increased until this deformation limit was reached at a concentrated load $R_{FE} = 5.84$ kN. According to Equation (5), the characteristic load-bearing capacity $R_k$ in the SLS was:

$$R_k = \frac{5.84}{1.01} = 5.78 \text{ kN} \tag{11}$$

### 3.4. Discussion

In ETA-14/0354 [5] for glulam made of **beech laminated veneer lumber GL75,** a **MOE in grain direction** $E_{L,mean} = 16{,}800$ N/mm² (MC = $5^{\pm3}$ %, $\rho \approx 800$ kg/m³) is specified. The experimentally determined mean MOE are in good agreement ($E_{L,mean} = 17{,}259$ N/mm², $E_{L,m,R,mean} = 17{,}667$ N/mm² and $E_{L,m,T,mean} = 17{,}576$ N/mm²; MC = 5.5%; $\rho \approx 803$ kg/m³; Table 1) and are 3 to 5% above the value of the ETA. The differences to $E_{L,mean} = 17{,}171$ N/mm² (MC = 7.1%, $\rho \approx 849$ kg/m³) assessed by Kuck [21] are even smaller with a maximum of 3%. Ehrhart and Steiger and Frangi [19,20] determined an up to 9% lower $E_{L,mean} = 16{,}156$ N/mm² (MC = 6.1%, $\rho \approx 796$ kg/m³).

In ETA-14/0354 [5], the **MOE perpendicular to the grain** is given as $E_{R/T,mean} = 470$ N/mm² (MC = $5^{\pm3}$ %, $\rho \approx 800$ kg/m³). The MOE determined here (Table 1) is twice as large with $E_{R,mean} = 840$ N/mm² and $E_{T,mean} = 966$ N/mm². In test report no. 176121 [17], higher values are also reported ($E_{R,mean} = 793$ N/mm², $E_{T,mean} = 786$ N/mm²; MC = 6.2%, $\rho \approx 822$ kg/m³). The value in the ETA is thus far on the safe side and might be adjusted to allow for a more economic design.

ETA-14/0354 [5] states a **shear modulus** $G_{LR/LT,mean} = 850$ N/mm² (MC = $5^{\pm3}$ %, $\rho \approx 800$ kg/m³). The experimentally determined mean shear moduli are in good agreement

($G_{\text{LR,mean}}$ = 909 N/mm², $G_{\text{LT,mean}}$ = 1006 N/mm²) and are 7% and 18% above the value of the ETA. The authors are not aware of any other published values for comparison. For the determination of the shear moduli, the method according to Albers [40] was used, which has proven to be well suited. This is also supported by the numerical comparative calculations (see Table 3). Further numerical investigations concerning different experimental methods for the determination of shear moduli of timber are currently being conducted and will be presented soon.

Up to now, the **Poisson's ratios** for glulam made of beech LVL were not known. The comparison with the results of Neuhaus [42] for spruce at MC = 4% (Table 5), shows that the presented experimentally determined values are in the same range. Higher differences in $\mu_{\text{RL}}$, $\mu_{\text{LT}}$ and $\mu_{\text{RT}}$ could be caused by the different macroscopic structure of softwood and hardwood and the production process of beech LVL GL75, which involves compression at high temperatures. The observation of an asymmetrical stiffness matrix by Neuhaus [42] can neither be confirmed nor rejected (Equation (8)). When measuring transverse strains with the optical measuring system Aramis, it should be considered that these strains lie within the range of the measuring accuracy/resolution limit for a measuring field of 300 mm × 220 mm (high COV of the Poisson's ratios, Table 1). The measurement noise can be compensated by averaging over many values or by a high measurement frequency and a subsequent regression analysis.

**Table 5.** Comparison of Poisson's ratios of spruce according to Neuhaus [42] for MC = 4% and own experiments on beech LVL GL75.

| | $\mu_{\text{LR}}$ [-] | $\mu_{\text{RL}}$ [-] | $\mu_{\text{LT}}$ [-] | $\mu_{\text{TL}}$ [-] | $\mu_{\text{RT}}$ [-] | $\mu_{\text{TR}}$ [-] |
|---|---|---|---|---|---|---|
| Neuhaus [42] | 0.2405 | 0.0505 | 0.2152 | 0.0291 | 0.3644 | 0.1964 |
| Experiments | 0.3127 | 0.0145 | 0.5167 | 0.0270 | 0.1978 | 0.2506 |

The determined elastic material stiffness matrix, Equation (8), can be used for numerical material modelling of beech LVL GL75 with a MC ≈ 5.5%. For different moisture contents in different service classes, the stiffness matrix should be adapted. Values for moisture content dependency of $E_{\text{L}}$ are given by Ehrhart and Steiger and Frangi [19]. Investigations of the moisture content dependency of MOE perpendicular to the grain, shear moduli and Poisson's ratios of beech LVL are not known to the authors.

With the presented experimental results, the knowledge gap concerning elastic material parameters of beech LVL is closed. This allows for more sophisticated numerical modelling of beech LVL and provides important findings for the guidelines.

It has been demonstrated that **optical measuring systems** such as Aramis are well suited for investigating material characteristics.

The procedure for the FE based determination of the resistance according to the *Guidelines for a Finite Element Based Design of Timber Structures* [1] is demonstrated. The model uncertainty is determined with the partial factor for modelling $\gamma_{\text{FE}}$ and taken into account in the design. The results of the FE model and the experimental investigations (serve as benchmark) are almost identical and $\gamma_{\text{FE}}$ = 1.01. The FE model is therefore very well suited for the numerical design of beech LVL beams. The described numerical model can be used as a benchmark for similar problems.

The demonstrated numerical design method opens the possibility to design timber structures, which are not covered by the design methods given in Eurocode 5 (standard design cases), within the safety concept of the Eurocodes by means of a numerical analysis. Furthermore, with the method "numerical simulation", it is possible to explicitly take into account material scattering in the numerical analysis, and, with the help of scattering input values and a Monte Carlo simulation, to determine the design resistance $f_{\text{d}}$ directly by means of a statistical evaluation according to EN 1990 [3].

## 4. Benchmark—Dowel-Type Connections for Beech LVL

### *4.1. General*

Within the IGF research project No. 20625 N [34] at the University of Stuttgart, tensile tests on steel-timber dowel-type connections in soft- and hardwoods were carried out to determine the connection stiffness. For the recalculation of the tests and to extend the parameter field by a numerical study, a Beam-on-Foundation (BoF) model was developed in RFEM (Dlubal). In addition, embedment tests were conducted within the research project RP 7-1 of the Cluster of Excellence IntCDC [2] to determine embedment curves, especially for dowels in beech LVL as input values. A detailed description of the investigations is given in [33,34]. The results of this benchmark case will be included in the guidelines and illustrate their application for FE modelling of dowel-type connections in beech LVL.

### *4.2. Experimental Investigations*

#### 4.2.1. Test Specimens, Configuration and Execution

In the scope of the IGF research project No. 20625 N (see [34]), a total of 260 component tests on steel-timber dowel-type connections were conducted with the aim of determining the load–deformation behaviour and in particular the initial stiffness. In addition, 18 tensile tests of fasteners were conducted, as the nominal values of steel can deviate significantly from the actual material properties. These results were needed as input values for the numerical model. The complete experimental programme is given in [29]. In addition to a large number of component tests in spruce glulam, a total of six specimens of beech LVL (GL75) with flatwise arranged veneer layers and a fastener diameter of 16 mm were tested, with the focus in this paper being on these dowel-type connections using beech LVL.

In order to increase the number of tested connections and thus the database, the tests were carried out symmetrically, so that two connections per specimen were tested at the same time. An example of the test setup as well as the specimen of the tensile test of a connection with three fasteners parallel to grain and a centred reinforcement is shown in Figure 8. The measurement devices were installed in conformity with EN 383 [44]. The load was applied statically according to EN 26891 [29,39].

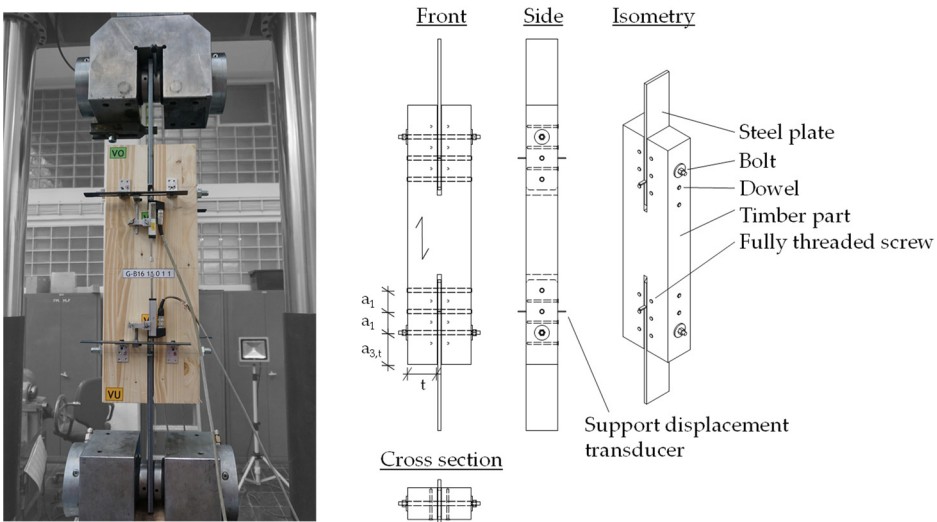

**Figure 8.** Example of the test setup for the tensile tests (**left**) [34], and of a specimen for tensile test parallel to grain with 1 × 3 fasteners (**right**) [29].

Additionally, within the framework of IntCDC, RP 7-1 [2], 93 embedment tests on softwood (GL 24h) and beech LVL specimens were conducted in conformity with EN 383 [44]. Own tests should ensure comparable boundary conditions of the component and the embedment tests, so that the results could be used as input values for the numerical model [29]. In the case of the specimens made of beech LVL, a distinction was made

between the edgewise- and flatwise-arranged veneers (Figure 9). As for the component tests, the diameter of the fasteners was always 16 mm. Further details on the embedment test, especially the ones using spruce glulam, are given in [29].

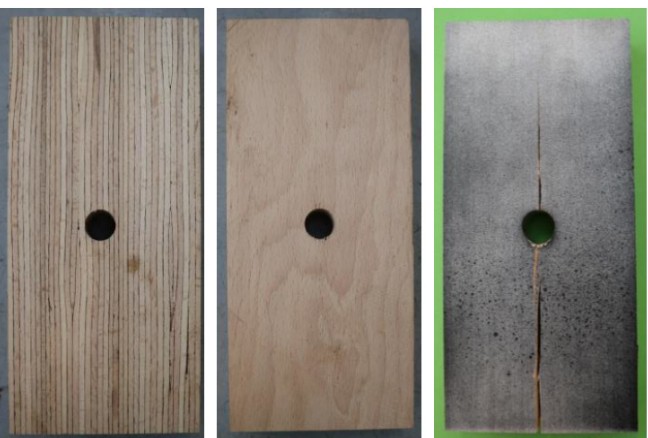

**Figure 9.** Beech LVL specimens for embedment tests with edgewise-arranged veneers (**left**), with flatwise-arranged veneers (**centre**) and with stochastic pattern after testing (**right**) [34].

Figure 10 shows the test setup for the embedment tests at a load-to-grain angle 0° and 90°. The load was applied to the fastener through a steel fork according to EN 26891 [39]. In this case, the deformation was measured using the optical measuring system Aramis Adjustable 4M and 12M by GOM GmbH. For this purpose, a stochastic pattern was sprayed onto the timber sample, which is shown on the right in Figure 9, and reference point markers were applied to the cross-section of the fastener.

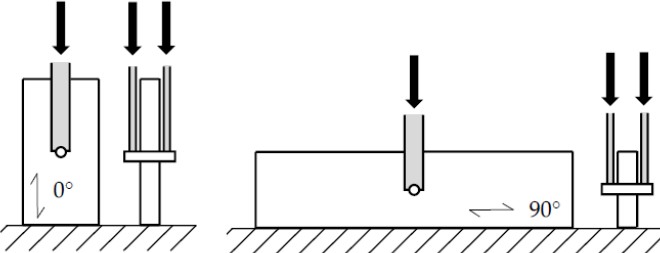

**Figure 10.** Test setup for the embedment tests at a load-to-grain angle 0° and 90° [34].

4.2.2. Evaluation and Results

The main test results for the initial stiffness $K_{\mathrm{ser}}$ and the reloading stiffness $K_{\mathrm{e}}$ from the tensile component tests on steel-timber dowel-type connections using beech LVL are summarised in Table 6 for the test series with and without reinforcement by means of fully threaded screws. In addition, a comparison of the mean values to the stiffness according to EN 1995-1-1 [26], $K_{\mathrm{EC5}}$ = 63.1 kN/mm, as well as the respective standard deviation (SD) and the COV was given. The stiffness specified in Eurocode 5 [26] was better represented by the experimentally determined initial stiffness $K_{\mathrm{ser}}$ than the reloading stiffness $K_{\mathrm{e}}$, which was underestimated. Furthermore, there was a slight influence of the reinforcement on the initial stiffness $K_{\mathrm{ser}}$. However, the large scatter with a COV of up to 28%, despite the homogeneous structure of beech LVL, and the small number of tests (six connections tested per series) did not allow for a clear conclusion about the load–deformation behaviour so far. For this reason, further tests on connections with beech LVL are necessary and are currently being conducted within the scope of a research project at the Institute of Structural Design [45].

**Table 6.** Initial stiffness $K_{ser}$ and reloading stiffness $K_e$ from tensile tests on steel-timber dowel-type connections using beech LVL, Ø = 16 mm [33,34].

| Series | Mean [kN/mm] ($K_{test}/K_{EC5}$) | | SD [kN/mm] | | COV [%] | |
|---|---|---|---|---|---|---|
| | $K_{ser}$ | $K_e$ | $K_{ser}$ | $K_e$ | $K_{ser}$ | $K_e$ |
| G-SD16 11 0 1_GL75 | 55.5 (88%) | 88.7 (141%) | 14.1 | 16.3 | 25.5 | 18.4 |
| G-SD16 11 0 2_GL75 | 68.0 (108%) | 108.3 (172%) | 19.0 | 30.8 | 28.0 | 28.4 |

Short name: "SD16 11 0 2" = dowel – Ø = 16 mm – 1 × 1 – 0°—no reinforcement (1)/centred reinforcement (2)—beech LVL (GL75); SD = standard deviation, COV = coefficient of variation.

The fasteners (dowels and bolts) were ordered with grade S235JR and were made of cold-drawn, galvanised bars. Using a randomly selected sample of the fasteners, the material properties were determined by means of tensile tests. The mean values of the yield strength $R_{p\,0.2}$, the tensile strength $R_m$ and the MOE are summarised in Table 7 for a fastener diameter of 16 mm. A comparison of the tensile strength actually determined with the minimum strength for S235 according to EN 1993-1-1 [46] shows a clear overstrength, which occurred in the same manner in other research projects (e.g., [27,29,30,32]).

**Table 7.** Mean values of the material properties of fasteners from tensile tests [29].

| Diameter [mm] | Yield Strength $R_{p\,0.2}$ [N/mm$^2$] | Tensile Strength $R_m$ [N/mm$^2$] | MOE [N/mm$^2$] | No. of Tests |
|---|---|---|---|---|
| 16 | 595.3 | 623.9 | 208,250 | 5 |

The failure of the embedment tests always occurred at a force-fibre angle of 0° due to the vertical splitting of the timber starting from the contact area of the fastener in the timber (Figure 9 (right)). Table 8 shows the main test results from the embedment tests using beech LVL at a force-fibre angle of 0°, which at the same time represent important parameters for the timber embedment properties approximated according to Equation (1) for the BoF model. The mean values as well as the standard deviation of the embedment parameters are given. The flatwise arrangement of the veneers corresponds to the setup of the steel-timber tensile tests shown in Table 6.

**Table 8.** Mean values of important parameters from embedment tests using beech LVL [33,34].

| Series | $k_{ser}$ (SD) [N/mm/mm$^2$] | $k_e$ (SD) [N/mm/mm$^2$] | $k_f$ (SD) [N/mm/mm$^2$] | $f_{h,int}$ (SD) [N/mm$^2$] | $u_0$ [mm] | $\alpha$ [-] |
|---|---|---|---|---|---|---|
| B160E | 144.8 (15.9) | 195.2 (16.3) | 1.5 (1.4) | 58.3 (4.7) | 0.18 | 9.3 |
| B160F | 149.8 (14.9) | 204.6 (13.7) | 0.54 (1.0) | 61.2 (2.4) | 0.15 | 7.8 |
| **Mean** | **147.3 (15.1)** | **199.9 (15.4)** | **1.02 (1.3)** | **59.8 (3.9)** | **0.17** | **8.6** |

SD = standard deviation; short name: "B160E(F)" = Beech LVL – Ø = 16 mm – 0°—Edgewise (Flatwise) veneers.

### 4.3. Numerical Investigations

#### 4.3.1. Modelling

In order to be able to investigate the influence of various parameters on the load–displacement behaviour of dowel-type connections in more detail (**step (1)** in Figure 1), a numerical model was developed by Kuhlmann and Gauß [33,34] using RFEM (Dlubal). The chosen model approach is based on a beam embedded on non-linear springs, which was first presented by Hochreiner et al. [31] and expanded by Schweigler [32]. Figure 11 shows the scheme of the Beam-on-Foundation (BoF) model and gives an overview of its individual elements, which are described in more detail below.

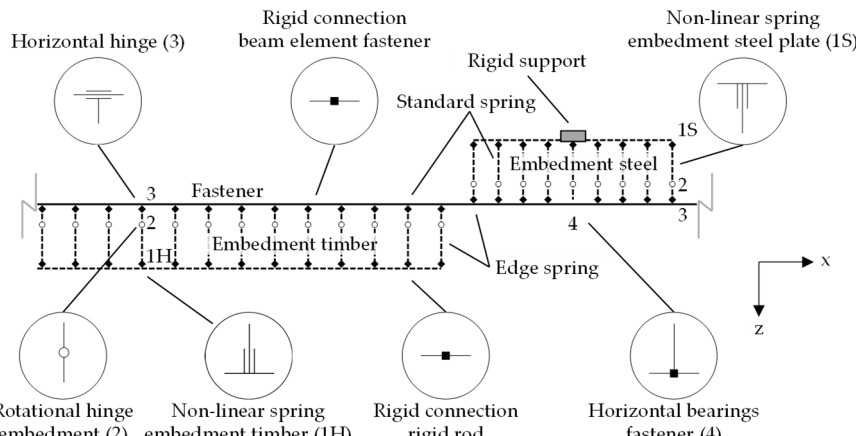

**Figure 11.** Details of the numerical Beam-on-Foundation (BoF) model and overview of the individual elements [29].

The FE model was more advanced than the mechanical models in EN 1995-1-1 [26]. However, no scattering input values, but average values were considered, as the design method **numerical design calculation with direct resistance check** was chosen (**step (2)**). An implementation of the model in common engineering software would therefore allow for a direct numerical computation of the connections' resistance and stiffness.

For numerical modelling of the tests (**step (3)**) according to the chosen design method, measured input values were taken for the geometry and the material parameters of the FE model for verification and validation (Section 4.3.2, **step (4)**). Due to the large material scatter in the tests, neither the model factor $\gamma_{FE}$ nor the design values were determined in this benchmark. The validation was conducted by quantifying the deviation of the model results from the test results by comparing the mean values.

The embedment properties of the timber represented the main input values and influenced the load–displacement behaviour of the connection most. In the model, the embedment properties were implemented on the basis of the experimentally determined embedment curves from the embedment tests (Table 8). Therefore, the test results were approximated by the parameterised curve according to Equation (1). In RFEM, this analytical approximation was described with a multilinear approach by nine individual points (Figure 12).

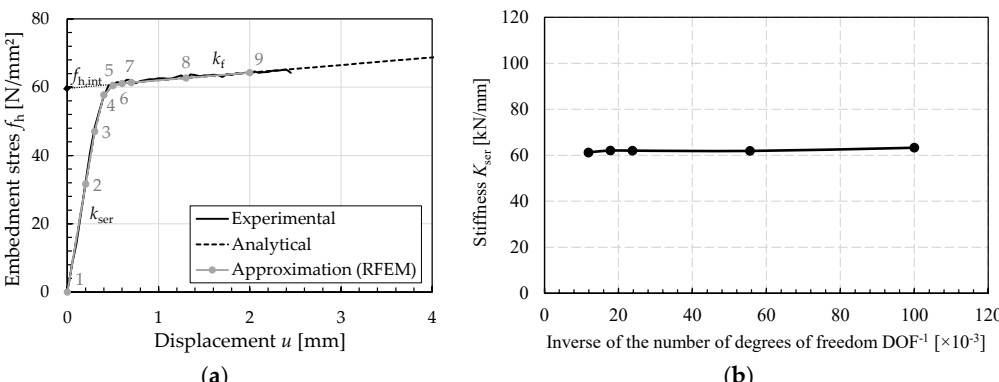

**Figure 12.** (**a**) Comparison of experimental embedment curve, analytical approximation and approximation by individual points [34]; (**b**) Numerically determined stiffness $K_{ser}$ when varying the number of embedment springs (plotted over the inverse of the degrees of freedom $DOF^{-1}$) for the component tests with beech LVL.

The modelling of the steel plate embedment followed the same principle as the modelling of the contact between the fastener and the timber and was described in RFEM by 12 individual points. However, the deformations occurring in the area of the slotted-in

steel plate could not directly be measured in the tests. Therefore, the required spring properties were determined by means of a 3D-FE model in Ansys Workbench 18.0 in order to be able to numerically implement the non-negligible flexibility in the area of the steel plate [33,34]. In addition, the contact area between the steel plate and the fastener was further developed compared to the initial model by Hochreiner et al. [31]. Through the arrangement of further non-linear springs, a precise description of the deformations of the fastener and the development of the plastic hinges at the edge of the steel plate was possible [29].

The fastener itself was approximated as a continuous beam element, supported on the non-linear springs, with a circular cross-section and non-linear material properties.

The material model of the fastener was also described by a multilinear approach with 12 individual points that represented the stress–strain curve of the previously conducted tensile tests on the fasteners [33,34].

### 4.3.2. Verification and Validation

The **verification** of the model was conducted analogous to *Guidelines for a Finite Element Based Design of Timber Structures* [1] (see Section 2.3).

1.  **Engineering judgement:** In Figure 13a the influence of the stiffness parameter $k_{\text{ser}}$ on the embedment stress in the timber is given. For smaller values of $k_{\text{ser}}$, the initial stiffness decreased. Especially in the range of small values, the parameter $k_{\text{ser}}$ had a significant influence on the embedment stress, while the influence in the range of the plastic plateau was negligible. Concerning the influence of the stiffness on the embedment stress, the model therefore showed a plausible behaviour.

2.  The **discretisation** was carried out depending on the number of non-linear springs which determined the number of finite elements. For this reason, the springs were evenly distributed. In addition, the side member thickness was a multiple of the selected spring spacing in order to avoid convergence problems. Figure 12b shows the numerically determined stiffness $K_{\text{ser}}$ for the component tests made of beech LVL when varying the number of embedment springs, plotted over the inverse of the degrees of freedom $\text{DOF}^{-1}$. Apparently, there was no significant influence of the discretisation on the connection stiffness.

3.  **Sensitivity check of input parameters:** As already described under point 1 "engineering judgement", the parameter $k_{\text{ser}}$ had a significant influence on the embedment stress, especially in the range of small values. The input parameter $k_{\text{ser}}$ should therefore be specified as accurately as possible, if the focus of the modelling is on connection stiffness. In contrast, the second stiffness parameter $k_{\text{f}}$ (plastic stiffness) only influenced the end gradient and maximum value of the slip curve (see Figure 13b). For modelling of the connection stiffness, this parameter was therefore of minor significance. The kink in the initial part of the curves in Figure 13a,b depends significantly on the parameter $u_0$ (initial slip). Since the stiffness was determined by the slope of the load–deformation curve between 10% and 40% of the maximum load, this kink did not influence the calculation of the stiffness. The sensitivity check of further input parameters is given in detail in Kuhlmann and Gauß [33,34].

4.  In this case, the **imperfection sensitivity analysis** can be omitted, as in connections with single dowels the influences of geometric imperfections, such as the dowel–hole clearance, can be neglected. However, an influence of the hole clearance on the load–displacement behaviour of fastener groups was given. First, investigations by Kuhlmann and Gauß [33,34] showed that the hole clearance between the fasteners and the slotted-in steel plate had a decisive influence on the stiffness of connections with fastener groups, since the stiffness cannot completely be developed due to the late activation of individual fasteners.

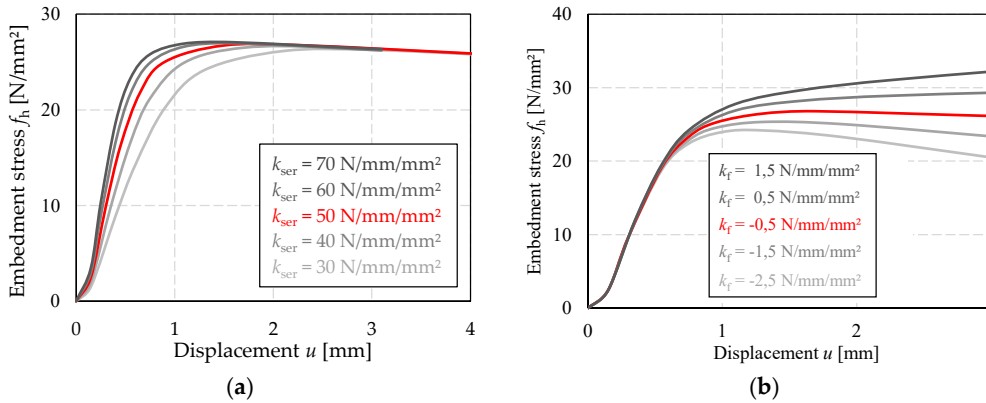

**Figure 13.** (**a**) Comparison of numerical embedment curves in timber for different values of $k_{ser}$ [34]; (**b**) Comparison of numerical embedment curves in timber for different values of $k_f$ [34].

With the verified numerical model, a recalculation of the embedment tests (see Figure 14) as well as of the 12 tested connections in beech LVL (see Figure 15) was carried out in order to **validate** the FE model. As input values for the embedment properties of the timber, the experimentally determined mean values of the embedment test series B160F were used to define the relevant parameters according to Table 8 (see also Figures 14 and 15, "Input values").

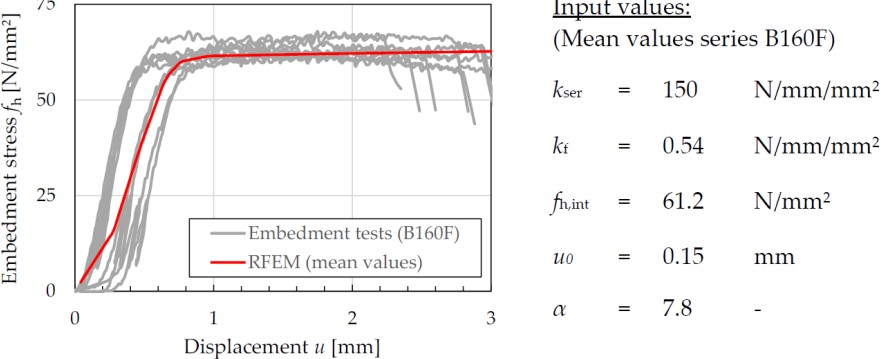

Input values:
(Mean values series B160F)

| | | | |
|---|---|---|---|
| $k_{ser}$ | = | 150 | N/mm/mm² |
| $k_f$ | = | 0.54 | N/mm/mm² |
| $f_{h,int}$ | = | 61.2 | N/mm² |
| $u_0$ | = | 0.15 | mm |
| $\alpha$ | = | 7.8 | - |

**Figure 14.** Experimental ("Embedment tests") and numerical ("RFEM") embedment curves of the test series B160F [34].

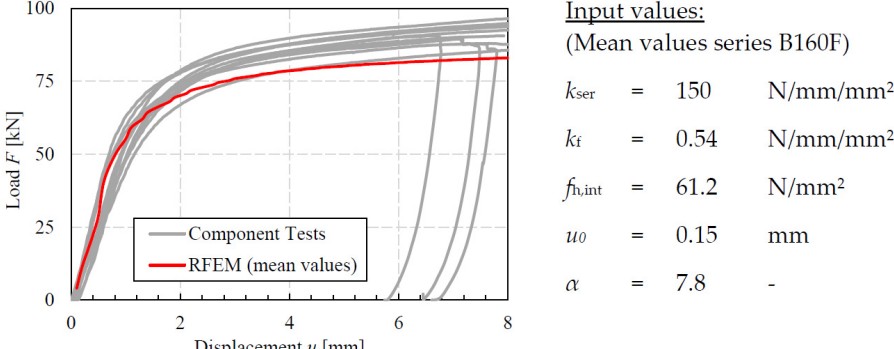

Input values:
(Mean values series B160F)

| | | | |
|---|---|---|---|
| $k_{ser}$ | = | 150 | N/mm/mm² |
| $k_f$ | = | 0.54 | N/mm/mm² |
| $f_{h,int}$ | = | 61.2 | N/mm² |
| $u_0$ | = | 0.15 | mm |
| $\alpha$ | = | 7.8 | - |

**Figure 15.** Experimental ("Component tests") and numerical ("RFEM") load–displacement curves of the test series with beech LVL [34].

Figure 14 compares the experimentally determined embedment curves (grey) with the numerically determined curve (red). According to Table 8, the tested initial stiffness was $k_{ser,test}$ = 149.8 N/mm/mm² for the series B160F. The corresponding numerically derived stiffness was about 30% lower with a value of $k_{ser,RFEM}$ = 114.3 N/mm/mm².

This deviation is not due to an inaccuracy of the model, but it can be explained by the chosen approximation of the timber embedment behaviour in RFEM using nine individual reference points [34]. A more precise description of the embedment behaviour with more than nine reference points could reduce this deviation. However, the numerical results are within the scatter range of the experiments and show a good accordance in the area of the initial stiffness as well as in the area of the plastic plateau.

Figure 15 compares the experimentally determined load–displacement curves (grey) of the test series with beech LVL with the numerically determined curve (red). The wavy shape of the numerical curve could be related to the settings of the global calculation parameters in RFEM, and here in particular to the selected number of iterations and the number of load increments. However, the further adjustment of these parameters to improve the wavy shape led to convergence problems. Nevertheless, the numerical results are within the scatter range of the experiments. Especially in the area of the initial stiffness, the numerical curve represents the experimental curves well. In the area of the plastic plateau, the numerically determined values tend to be in the lower range compared to the test results. The numerically derived initial stiffness $K_{check}$ was calculated using the straight-line slope and was thus $K_{check}$ = 61.2 kN/mm. According to Equations (3) and (4), across all tests the mean value of the ratio $K_{test,known}/K_{check}$ was $m_x$ = 1.01 and the coefficient of variation $V_x$ = 0.28. This shows, on the one hand, the good agreement of the experimentally and numerically determined mean values of the initial stiffness. On the other hand, the high coefficient of variation also demonstrates the large scatter of the experimental results. For this benchmark, due to this large scatter, a calculation of the $\gamma_{FE}$ factor according to Equation (2) was not appropriate and therefore not given here.

*4.4. Discussion*

In this case study, the experimental and numerical investigations of Kuhlmann and Gauß on tensile tests of steel-timber dowel-type connections with beech LVL in the frame of the IGF research project No. 20625 N [34] as well as embedment tests using beech LVL within the Cluster of Excellence IntCDC [2] have been presented. The aim of the research projects was to identify and quantify further influencing factors on the connection stiffness, as currently only the density of the timber and the diameter of the fastener are considered in EN 1995-1-1 [26]. A detailed description of the results regarding the influencing factors is given in [29,33,34]. Furthermore, it was examined to what extent the normative rules are also applicable for connections in hardwood. The results of the **tensile tests on steel-timber dowel-type connections** show that the initial stiffness $K_{ser}$ seems to be better represented by the standard than the reloading stiffness $K_e$, which was rather underestimated by the values according to the code (see Table 6). However, despite the homogeneous structure of beech LVL, a large scatter with coefficients of variation of up to 28% is determined. The small number of the additional six tested connections per series do not allow for a clear conclusion about the load–deformation behaviour so far. Further tests on connections with beech LVL are currently being conducted within the scope of a current research project at the Institute of Structural Design [45].

Based on the experimental investigations, a **Beam-on-Foundation (BoF) model** of the component tests was developed in RFEM (Dlubal) in order to investigate the load–deformation behaviour and the influencing factors more intensively. The results of these numerical investigations will be included in the guidelines. A detailed parameter study on the influences on the connection stiffness was carried out by Kuhlmann and Gauß [33,34]. The implemented numerical model is based on a beam, representing the fastener, embedded on non-linear springs, which describes the embedment behaviour of the fastener in the timber as well as in the steel plate.

The mentioned **embedment tests on beech LVL specimens** were carried out in order to have realistic input values for the numerical model and to ensure comparable boundary conditions for the component tests at the same time. Based on the experimental embedment curves, several important parameters according to the analytical approach by Schwei-

gler [32] (Equation (1)) were determined (see Table 8). In a further step, the embedment behaviour was implemented in RFEM by means of an approximation of the analytical solution using nine individual reference points. The timber embedment behaviour was found to have the greatest influence on the numerically derived load–deformation curves.

In addition to the embedment tests, **tensile tests** on randomly selected samples **of the fasteners** were carried out as well (see Table 7). A comparison of the actual tensile strength with the minimum nominal strength of S235 according to EN 1993-1-1 [46] shows a clear overstrength, which occurred in the same manner in other research projects (e.g., [27,30,32]).

According to the *Guidelines for a Finite Element Based Design of Timber Structures* [1], the BoF models of the embedment tests and the component tests were first verified and then validated by comparison with the test results. Experimentally determined mean values were used for the input parameters of the embedment properties of the timber in the numerical model. Although the numerically derived stiffness of the embedment tests was about 30% lower than the experimentally determined stiffness, the numerical results were within the scatter range of the experiments. This large scatter was also evident in the experimentally derived initial stiffnesses of the tensile tests on steel-timber dowel-type connections and can probably be attributed to material scatter. However, for a comparison of the numerical and experimental results across all tests, the mean value of the ratio $K_{\text{test,known}}/K_{\text{check}}$ was $m_\text{x} = 1.01$ and thus showed a good agreement in mean. Nevertheless, the material scatter should be taken into account in the numerical model for the FE based design. The calculation of a $\gamma_{\text{FE}}$ factor did not seem appropriate for this benchmark. For dealing with the large scatter, a numerical simulation (see Figure 1), which directly considers the scatter within the safety evaluation, might be more suitable.

## 5. Conclusions and Outlook

Within the Cluster of Excellence IntCDC at the University of Stuttgart, *Guidelines for a Finite Element Based Design of Timber Structures* [1] have been developed at the Institute of Structural Design and are presented in this paper. The guidelines allow for a design of timber structures beyond the standard design cases of the Eurocodes and enable structural engineers to design innovative constructions and to verify them within the framework of the Eurocodes using the finite element method (FEM). Three different design methods for a finite element based design of timber structures are presented depending on the design purpose:

- Numerical design calculations requiring a subsequent design check: for the daily engineering practice application of FEM for standard design cases covered by the Eurocodes.
- Numerical design calculations with direct resistance check: for the expert engineering application of FEM, e.g., for design cases beyond the standard design cases of the Eurocodes.
- Numerical simulations: for the use of numerical methods for the complementation, extension or replacement of physical experiments, e.g., for product development and certification.

In addition, guidance for the application of FEM in research is given.

The guidelines are intended to stimulate discussions on the use of FEM in timber engineering. In addition, the possibilities of FEM in design are raised to a new level, namely, the direct determination of resistances (not only internal forces and deformations) considering the safety concept of the Eurocodes. Finally, the guidelines are meant to provide an impetus for the standardisation process in timber construction, similar to prEN 1993-1-14 [4] for steel structures.

Two **benchmarks** illustrate the procedure for a FE based design and describe the modelling, verification, validation and design using numerical methods. The benchmarks are supported by experimental investigations, in which the elastic material behaviour and the stiffness of dowel-type connections in beech LVL GL75 (ETA-14/0354 [5]) were investigated. Additionally, the experimental investigations provide important input values for the numerical modelling of beech LVL, and their results are included in the guidelines.

In the experimental investigations of the benchmark on the **elastic material behaviour of beech LVL**, the complete elastic material stiffness matrix of beech LVL was determined for the first time. For this purpose, surface strains and deformations were measured with the optical measuring system Aramis, and, in addition to the moduli of elasticity (MOE) and shear moduli, all six Poisson's ratios were determined. Equation (8) can be used for numerical modelling of the elastic material behaviour beech LVL.

Based on the experimental results, it can be concluded that the MOE perpendicular to the grain $E_{R/T,mean}$ = 470 N/mm$^2$ given in ETA-14/0354 [5] seems to be on the safe side and might be adjusted to a value of $E_{R/T,mean}$ = 800 N/mm$^2$ to allow for a more economic design. The other experimentally determined MOE and shear moduli agree well with the values in ETA-14/0354 [5].

Further investigations of the elastic material properties of beech LVL should include the influence of the wood moisture content, the shear modulus $G_{RT}$ and a more precise determination of the Poisson's ratios.

For the benchmark on **dowel-type connections in beech LVL**, embedment and component tests were conducted. The initial stiffness $K_{ser}$, the reloading stiffness $K_e$ (Table 6) and the embedment parameters (Table 8) were computed. The applicability of the design rules of EN 1995-1-1 [26] for hardwoods was investigated (Table 6). It was shown that the initial stiffness $K_{ser}$ was better represented by the standard than the reloading stiffness $K_e$, which was rather underestimated by the values according to the code. However, there was a large scatter with a COV of up to 28%. Thus, further extensive test series are necessary and currently being conducted in a research project [45].

**Author Contributions:** Conceptualisation, J.T., L.B. and U.K.; resources, U.K.; guidelines FE based design and their description, J.T.; elastic material behaviour, conduction of experiments, J.T. supported by J.L.; evaluation, J.L.; FE modelling, J.L. and J.T.; writing—original draft preparation, J.L. and J.T.; FE modelling, L.B.; writing—original draft preparation, L.B.; writing—review and editing, U.K. and J.T.; visualisation, J.T., L.B. and J.L.; supervision, U.K.; project administration, AiF, IntCDC, J.T.; funding acquisition, AiF. and U.K., IntCDC, J.T. and U.K. All authors have read and agreed to the published version of the manuscript.

**Funding:** The work on the FE guidelines, the elastic material tests and the embedment tests were supported by the Deutsche Forschungsgemeinschaft (DFG, German Research Foundation) under Germany´s Excellence Strategy—EXC 2120/1—390831618. The component tests were supported by the Federal Ministry for Economic Affairs and Energy on the basis of a decision of the German Bundestag (IGF project No. 20625 N).

**Acknowledgments:** We would like to give a big "Thank you" to the Material Testing Institute of the University of Stuttgart for the milling of test specimens, conduction and advising on the experiments. The support of Pollmeier Massivholz GmbH and Co. KG for the provision of the beech LVL test specimens is gratefully acknowledged. We thank our former colleague Julius Gauß (Contributions: dowel-type connections, conduction of experiments, evaluation, FE modelling, visualisation, project administration and funding acquisition) and our experimental technician Elisabeth Bokesch for their great and enduring work.

**Conflicts of Interest:** The authors declare no conflict of interest.

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
