# Peer review of "Guidelines for a Finite Element Based Design of Timber Structures and Their Exemplary Application on Modelling of Beech LVL"

_buildings, doi:10.3390/buildings13020393_

Round 1

Reviewer 1 Report

Thanks for this very interesting and well written manuscript. It is a very important contribution to the engineering community!

There are few things that need clarification in order to make the manuscript ready for publication.

General:

In the first example you perform the design in SLS with regard to deflection in midspan.

Please mention in the paper, how a design in ULS would look like, especially with regard to the following aspect: in the FEM model with the longitudinal stresses you show the highest bending stresses in tension on the lower side of the beam. However, in compression there are some stress concentrations with even higher stresses just at the load introduction point. How should these higher compression stresses be interpreted?

An experienced engineer might compare the risk of brittle bending tension and ductile bending compression failure and focus on the brittle failure.

How does an un-experienced engineer deal with that situation or how do you approach the problem in a more complex situation? How do you identify the relevant failure mode in ULS?

How to deal with the local stress concentration from the load introduction? In more complex structures this identification of the relevant failure mode might not be as simple as for a beam in 3 point bending.

Please give a brief explanation on how to deal with this problem!

Line 185

Please consider and clarify that the \gamma_M in EC5 does (besides material variability) only account for a “general” model uncertainty for the reference case this factor is calibrated on. In special cases such as connections, notches, curved beams etc (and in particular when FEM is used) other and possible higher model uncertainties might be necessary!

Line 403

Please critically review the number of significant digits, especially for the COVs! 3 digits might be maybe reasonable.

Line 494

“l_s” italic

Line 509 and 518

It is not clear which value E_L is given in the ETA, is it 15300 or 16800? Please clarify your choice!

Line 621

These lines may be a repetition from the first example and might be skipped here.

Line 633

It is not clear what you mean with “standard”. Do you mean the test standard or EC5? Please clarify e.g. “The stiffness specified in EC5 is better represented by the experimentally determined initial stiffness K_ser than the reloading stiffness K_e”.

Line 730

Please define the meaning of k_f!

Line 749 following

On both figures on embedment and connection behaviour, there is a kink in the initial part of the curve. Please explain where this kink comes from (initial slip?) and how it influences the calculation and interpretation of the stiffness!

Line 754

Please explain what need to be changed regarding reference points in order to achieve a higher value of k_ser,RFEM in the FEM model!

Line 758

Please also comment on the wavy shape of the numerical curve!

Line 816

Which “numerical results” do you refer to here? The stiffness results? Or the load-deformation curve or maximum load?

Author Response

Thank you very much for the important remarks and questions. We adapted the paper accordingly.

Concerning your general questions at the beginning:

These are very important questions, which are also covered in the FE guidelines. As an appropriate explanation would significantly increase the length of this article, the authors prefer not to discuss this topic here but separately in another publication. However, certain additional explanations are given in 2.2 and in principle the ULS design verification follows the same procedure as demonstrated for SLS.

A short summary of ULS design verification with FE according to the guidelines:

In case of numerical design calculations requiring a subsequent design check, internal forces are calculated and the formulas in EN 1995-1-1 are used for design verification. No direct stress verification is conducted at the FE model. This procedure is recommended for unexperienced engineers and the standard design cases described by EC5.

In case of numerical design verifications with direct resistance check, numerically determined stresses and strains are directly compared with limit stresses and strains. Stress or strain singularities have to be treated separately (see guidelines). Limit strains are only relevant in compression, due to the elastic material behaviour of wood in shear and tension.

Limit stresses should be chosen according to EC5. Where no interaction rules are given in EC5, linear interaction may be assumed. As the manual checking of limit stresses of stress interactions can be quite cumbersome, this should be directly implemented in the material model or automatized in the postprocessing of the respective FE software.

It is suggested to limit compression strains to epsilon_el+pl,2,0 (see attached pdf). epsilon_el+pl,2,0 depends on the chosen material model (see attached pdf, e.g. bilinear, multilinear, …). This strain limit still has to be discussed in the timber community, as its exceedance does not directly lead to member failure.

This procedure can be used by experienced engineers, also for design cases beyond standard cases described in EC5.

In case of numerical simulations limit stresses and strains may be treated differently.

For ULS design verification of this benchmark in case of a numerical design verification with direct resistance check the authors suggest following checks:

- tension stress parallel to the grain (prevailing at the lower edge at midspan)

- compression stress and strain parallel to the grain (prevailing at the upper edge at midspan underneath load introduction). It still has to be discussed whether these limits may be omitted in special cases, as this does not lead to a member failure.

- compression stress and strain perpendicular to the grain (prevailing at the upper edge at midspan underneath load introduction or at the supports). The experienced engineer may omit this limit criteria, if the additional deformation does not significantly change the internal forces (e.g. in a truss).

- compression stress interaction (prevailing at the upper edge at midspan underneath load introduction)

- shear stress close to the supports

- stress and strain singularities at load introduction and supports can omitted

These checks should be directly implemented in the material model or automatized in the postprocessing of the respective FE software.

For determination of the relevant failure modes the engineer still depends on experience which can be supported by software internal checks of failure criteria. Singularities can be treated by the engineer according to Annex B of the guidelines.

For more complex structures, it is the opinion of the authors that software internal checks of failure criteria are necessary for supporting the engineers’ experience.

Concerning your question to line 185: in case of Numerical design calculation requiring a subsequent design check (according to EN 1995) standard cases are investigated which usually do not need an extra gamma_FE for model uncertainty. In case of numerical design calculation with direct resistance check this additional model uncertainty is dealt by gamma_FE. More explanations are given in the text.

Concerning your questions to line 403 and 494: changes done.

Concerning your question to line 509 and 518:

Both the characteristic and mean value of E_L are given in the ETA. Characteristic is used for SLS design and mean for comparison with experimental results.

Concerning your questions to line 621 and 633: changes done.

Concerning your questions to line 730 and 749: changes done.

Concerning your questions to line 754 and 758: changes done.

Concerning your questions to line 816: changes done.

Reviewer 2 Report

The paper is interesting, and the research content has a particular engineering application significance. In this paper, based on two experiments, the constitutive relation of beech veneer laminated lumber is described, on the basis of which a numerical model is established by the finite element method.

First of all, the manuscript seems to be compiled from excerpts of extensive research reports, and the full text does not have a good logical relationship. The above can be seen from the citation order of the references, such as line 41 on page 1, and the number of the first reference of the article is [39], which is obviously not in line with the format of academic paper writing. It is suggested that the author reorganize the structure of the full text.

At the same time, the conclusion of the article is too lengthy, so it is suggested to simplify it.

Author Response

Thank you very much for the important remarks.

We strengthened the logical relationship between the different parts, e.g. line 58 and 315.

The order of citation was adapted and the conclusions of the benchmarks in section 5 have been shortened.

Reviewer 3 Report

Overall, the presented manuscript "Guidelines for a Finite Element Based Design of Timber Structures and their Exemplary Application on Modeling of Beech LVL" is innovative and contributes significantly to increasing knowledge in the field and the practical applicability of finite element-based design on modelling of LVL. In my opinion, the article would be of considerable interest to the journal readers.

Therefore, I have only a few recommendations for the esteemed authors.

First - please check the journal's requirements for arrangement and citation in the text of the references. They are currently listed alphabetically rather than by citation order; please correct this.

Figure 1 (between lines 192-193) - please consider removing "according to" and leaving only the cited reference.

Line 320 – there is an error in the term: "moduli of elasticity", and please correct it to "modulus of elasticity". That also applies to table 1 (lines 403-404). It would be good to give the generally accepted abbreviations for bending strength - MOR and modulus of elasticity (MOE).

Please give axes titles in Figures 5 and 6 (between lines 370 and 371). In my opinion, this will significantly add to their clarity.

In line 379 – Why "results" is in bold?

Lines 426-430 – I ask the authors to consider whether using bold is necessary. To avoid using double parentheses, I suggest separating steps (2) and (3) with a comma.

Lines 453-458 – The previous recommendation applies to this paragraph: Figure 1 and step (4). Please reconsider whether using bold is necessary, which also applies to the overall text of the manuscript.

Figures 10 and 11 - I ask the authors to give axes titles for the horizontal axis.

Line 516, point 3.4. "Discussion" - in general, despite the in-depth discussion on the results obtained, this part lacks a comparative analysis with results achieved in similar studies.

Figures 18, 19, 20, 21, and 22 – My recommendation is again to be given axes titles.

The references cited are appropriate.

Author Response

Thank you very much for the important remarks and questions. We adapted the paper accordingly.

Concerning your comments to Figure 1 and line 320: changes done.

Concerning your comments to Figures 5 and 6: changes done.

The bold highlighting in line 358 to 458 is intended to highlight the substructures, which simplifies the readability. No change suggested.

Concerning your comments to Figures 10 and 11: changes done.

Concerning Line 516, point 3.4. "Discussion":

Results by Ehrhardt & Steiger & Frangi were added. No further published values are known to the authors. Many results of the certification tests of beech LVL are unfortunately not published. This in now stated at line 117.

Concerning your comments to Figures 18 to 22: changes done.

Round 2

Reviewer 1 Report

Thanks for considering my comments.